# Hypertension is the Prominent Risk Factor in Cataract Patients

**DOI:** 10.3390/medicina55080430

**Published:** 2019-08-02

**Authors:** Ioanna Mylona, Maria Dermenoudi, Nikolaos Ziakas, Ioannis Tsinopoulos

**Affiliations:** 2nd Department of Ophthalmology, Aristotle University of Thessaloniki, 564 29 Thessaloniki, Greece

**Keywords:** cataract, arterial hypertension, diabetes mellitus, dyslipidemia

## Abstract

*Background and objectives:* The purpose of this study is to determine the impact of the most prominent cardiovascular and metabolic risk factors in patients undergoing cataract surgery. *Materials and Methods:* The study included 812 consecutive patients undergoing unilateral, uneventful cataract surgery by means of phacoemulsification, at the 2nd Department of Ophthalmology, Medical School, Aristotle University of Thessaloniki, Greece, during a calendar year. Patients were assessed for the type of cataract and the presence of three diseases, under pharmacological treatment, that have been reported as risk factors for the development of cataract (arterial hypertension, diabetes mellitus, and dyslipidemia). *Results:* There was a statistically significant difference between the types of cataract and individual risk factors (*p* < 0.001). Hypertension was the most frequentrisk factor, ranging from 43.8% in patients with subcapsular cataracts, 24.3% in patients with nuclear cataracts, 28.6% in patients with cortical cataracts, and 27.6% in patients with mixed type cataracts. There was a statistically significant difference as to the total number of risk factors per cataract type (*p* < 0.001); almost all patients with subcapsular cataracts had at least one risk factor (98.4%) while this percentage was 90.5% for patients with mixed cataracts, 85.7% for patients with cortical cataracts, and78.6% for patients with nuclear cataracts. *Conclusions:* Diabetes mellitus did not have a large incidence in our sample as a single risk factor, while hypertension did. This finding raises the importance of early detection of hypertension, a cardiovascular condition that typically progresses undetected for a number of years.

## 1. Introduction

A cataract is a major cause of blindness in developed and developing countries. Various cardiovascular and metabolic conditions have been proposed as possible contributors to the etiopathology of the disease, including diabetes mellitus (DM), arterial hypertension, and dyslipidemia, since adequate control of those parameters have been shown to be beneficial to prevent cataract development and decrease its progression rate [1,2]. The incidence and progression rate of cataract is found to be elevated in diabetic patients, who are also at a higher risk of intra- and postoperative complications regarding cataract surgery compared to non-diabetics [3]. A recent meta-analysis concluded that arterial hypertension increases the risk of cataract, especially the posterior subcapsular subtype [4]. Literature data for dyslipidemia have been inconclusive to date. A study including 3654 elderly Australians revealed no significant association between baseline serum lipids or fibrinogen and incident cataract or cataract surgery [5]. Another large-scale study of 3251 Chinese subjects also did not find any link of dyslipidemia to nuclear, cortical, or subcapsular cataract [6]. However, both studies were population surveys and not studies of a clinical population. Nevertheless, dyslipidemia has been indirectly associated with cataract development, since statin use in a general population appears to be associated with a lower risk of nuclear cataract formation [7]. A study of 2794 Malay adults found that cataract prevalence increased with higher quartiles of blood glucose, systolic blood pressure (BP), and metabolic syndrome components while high BP was associated with all three cataract types. Diabetes was associated with cortical and posterior subcapsular (PSC) and low high-density lipoprotein (HDL), high body mass index (BMI), and metabolic syndrome were associated with cortical cataract [8]. The presence of both high BP and diabetes was associated with a four-fold increasein having cataract [8].

In this study, we sought to assess the relative incidence of the metabolic conditions in patients who underwent phacoemulsification surgery for all subtypes of senile cataract.

## 2. Materials and Methods

### 2.1. Subjects and Measurement

This is a cross-sectional study of all consecutive patients who underwent surgery for senile cataract by means of phacoemulsification at the 2nd Department of Ophthalmology, Medical School, Aristotle University of Thessaloniki, Greece, from March 2018 to April 2019. All examinations and surgical procedures were performed in the same Department. Institutional Review Board approval was obtained for this study (ref. ID 3877-3/ date of approval 06-07-2017) and all procedures adhered to the tenets of the Declaration of Helsinki. All patients included in the study were over 18 years of age and in anappropriate mental state to understand and sign the informed consent form.

As per our departmental policy, besides a full ophthalmic examination, a thorough medical history was obtained from each patient during their preoperative visit. Eight-hundred and fifty-six patients receiving phacoemulsification surgery were initially enrolled in the study. The inclusion criteria were the presence of clinically significant senile cataract with an indication for surgery. The ophthalmologic examination in all patients included best-corrected visual acuity (BCVA), applanation tonometry, slit-lamp examination, and fundus examination through dilated pupils. BCVA was measured with the Early Treatment Diabetic Retinopathy Study (ETDRS) charts. Measurement of contrast sensitivity (CS) was tested with a Pelli–Robson chart. Measurement of the central macular thickness (CMT) was determined using optical coherence tomography. Cataract assessment was performed using the slit lamp with a pharmacologically-induced pupil dilation of at least 6 mm. Initial grading was performed during the scheduled first outpatient appointment by both an ophthalmology resident and an ophthalmology consultant. An independent expert grader (last author) whose opinion prevailed in case of inter-rater disagreement confirmed the rating for the purpose of the study. The cataract type classification was performed according to the Lens Opacities Classification System III (LOCS III), a standard system used for grading and comparison of cataract type and severity [9].Grading is made with respect to nuclear opalescence (NO: 0.1–6.9), nuclear color (NC: 0.1–6.9), cortical (C: 0.1–5.9), and posterior subcapsular (P: 0.1–5.9) cataracts, while mixed cataract cases were patients with a coexistence of different pure lens opacities in one or both lenses.

Patients with traumatic or juvenile cataract, as well as cases with ophthalmic comorbidities which could lead to cataract formation (such as chronic uveitis, prior intraocular surgery, prolonged use of topical steroids) were excluded from the analysis. Patients receiving bilateral cataract surgery during the study period were included only for their chronologicallyfirst procedure. As such, 834 patientswere ultimately included in the study.

Of all medical conditions recorded in the patients’ medical history, the study focused on the presence of three main pathologies, which were specifically recorded for each patient. These conditions included Diabetes Mellitus (DM), Arterial Hypertension (AH), and Dyslipidemia (Dysl). Each patient was classified as having one or more of the above-mentioned conditions only when they had undergone a recent (<6 months) medical examination and they were receiving medical treatment for this specific condition.

### 2.2. Statistical Analysis

Scale variables were checked for normality using the Kolmogorov–Smirnov criterion and appropriate test statistics were employed. Group comparisons on categorical variables were calculated with Pearson’s Chi-square statistics. The non-normal continuous variables were analyzed with the Kruskal–Wallis test statisticsand the individual subgroup differences were compared by means of the Mann–Whitney tests, corrected with the Bonferroni criterion.

All group comparisons were computed with the IBM SPSS statistical package (version 25; IBM Corp., Armonk, NY, USA).

## 3. Results

Our research sample consisted of 454 female patients (mean age = 75.38 years, standard deviation (SD) = 6.95) and 380 males (mean age = 74.68 years, SD = 7.41).There was no statistically significant difference in age between genders (Mann–Whitney Z = 1.146, *p* = 0.252). The classification of cataract subtypes in each gender is provided according to the respective comparisons in Table 1.

There was no statistically significant difference between the groups in age, Pearson’s Chi-square = 5.933, df = 6, *p* = 0.431. The relative percentages are presented in Table 2.

Table 3 presents the results of the ophthalmological examination by types of cataract, there were no statistically significant differences between the groups in BCVA and CMT measurements, while cortical cataracts were more affected than the other types in CS (*p* < 0.001).

Table 4 presents the total number of risk factors by type of cataract (since most patients had at least one or more risk factors).

Patients with different types of cataract also had different frequencies of risk factors, and that difference was statistically significant (Pearson’s Chi-square χ^2^ (21) = 87.899, *p* < 0.001 while Cramer’s V = 0.187, *p* < 0.001). Patients with subcapsular cataracts typically presented either with hypertension (43.8%) or with hypertension and dyslipidemia (26.6%). Patients with nuclear cataracts typically presented either with hypertension (24.3%) or without risk factors (21.4%). Patients with cortical cataracts typically presented either with hypertension (28.6%) or with all three risk factors (21.4%). Finally, patients with mixed type cataracts typically presented either with hypertension (27.6%) or with hypertension and dyslipidemia (22.9%).

Table 5 presents the median, range and standard deviation of risk factors in each patient cataract group.

Cataract groups were compared on their total number of risk factors with the Kruskal–Wallis group statistical comparison for non-parametric variables, and were found to be significantly different to one another, Chi-square (3) = 18.436, *p* < 0.001. This finding does not explain which groups differedfrom one another, so thecomparisons between each group were madeusing the Mann–Whitney test. To correct for multiple comparisons according to the Bonferroni correction, the target *p*-value for an alpha of 0.05 (the minimum threshold for statistical significance) was set to 0.008. Results are presented in Table 6 and indicate that patients with subcapsular and mixed type cataracts had more risk factors on average than patients with nuclear cataracts (*p* values were <0.001 and 0.003, respectively). Effect sizes (d) for those differences were 0.439 and 0.266, respectively, denoting small to medium clinical significance. The effect sizes for the non-significant comparisons were smaller than 0.1 denoting that increasing the sample size for the study would be unwarranted.

Table 7 presents the number of patients per cataract type with at least one risk factor and those without any risk factors. There is a statistically significant difference between patients with different cataract types (Pearson’s Chi-square χ^2^ (3) = 36.833, *p* < 0.001). Almost all patients with subcapsular cataracts had at least one risk factor while this incidence dropped to 90.5% for patients with mixed cataracts, 85.7% for patients with cortical cataracts and 78.6% for patients with nuclear cataracts.

## 4. Discussion

This study replicates the findings of other studies by confirming that cataract patients present with high incidences of hypertension, diabetes mellitus, and dyslipidemia. The co-existence of multiple cardiovascular and metabolic risk factors has an additive effect on cataract etiopathology, particularly related to the metabolic syndrome, a loosely-defined clinical syndrome characterized by the presence of ≥3 of the following components: body mass index (BMI) ≥25 kg/m^2^, triglycerides ≥1.7 mM, high density–lipoprotein (HDL) cholesterol <1.0 mM in men and <1.3 mM in women, blood pressure (BP) ≥130/85 mm Hg, or use of BP medication and diabetes mellitus. Research on the relationship between individual cataract types and these metabolic conditions (AH, DM, Dysl) has been carried out on clinical [10] and non-clinical populations [11]. A case-control study of 385 cases and 215 controls in Italy concluded the risk factors for cortical cataract were the presence of diabetes for more than five years and increased serum K^+^ and Na^+^ levels [10]. In that study, posterior subcapsular cataract was associated with the use of steroids and diabetes, and nuclear cataract with calcitonin and milk intake whilemixed cataract was associated with a history of surgery under general anesthesia. A Korean large-scale epidemiological study of 11,591 participants [11] found that older age, lower monthly household income, lower education, hypercholesterolemia, and DM were independent risk factors for the development of pure cortical cataracts. In this study, older age, lower education, metabolic syndrome, and DM were independent risk factors for the development of pure nuclear cataracts. Further, older age and DM were independent risk factors for the development of pure posterior subcapsular cataracts. That study concluded that older age, lower monthly household income, lower education, and DM were independent risk factors for the development of mixed cataracts.

Although DM, AH and dyslipidemia may be viewed as comorbid conditions due to the increased age of the patients, the fact that there were significant differences between different types of cataract and the incidence of individual risk factors provides evidence to the premise that different risk factors are linked to different paths in cataract pathophysiology.

Considering our results, the most frequent risk factor for all types of cataract was hypertension. The incidence of patients presenting only with hypertension ranged from 43.8% in cases with subcapsular cataract to 24.3% in cases with nuclear cataract. The combination of hypertension with dyslipidemia also had high incidences among all the cataract groups, being the second most frequent in cases of subcapsular (26.6%) and mixed type (22.9%) cataracts while hypertension with both dyslipidemia and diabetes mellitus was also relatively frequent, ranging from 21.9% in subcapsular, 21.4% in cortical, 18.1% in mixed type, and 15% in nuclear cataracts. Interestingly, diabetes mellitus alone or in combination with dyslipidemia had very low incidences in all groups; there weren’t any cases of cortical cataracts, there were 1.6% of cases with subcapsular cataract, 6.8% of nuclear cataracts, and 9.5% of cases with mixed type cataract. This is an unexpected finding; although we may argue that the incidence of uncomplicated diabetes mellitus could generally be low in the population at large compared to diabetes mellitus with comorbid arterial hypertension or dyslipidemia. There is a possibility that hypertension may exacerbate the negative impact of diabetes mellitus in the progression of diabetic cataract. Initially in diabetes mellitus, the high level of glucose in the aqueous humor diffuses into the lens and is metabolized to advanced glycation endproductswhich accumulate within the lens and play an important role in cataract formation [12]. Intraocular pressure and systemic blood pressure have been found to correlate [13]. Increased intraocular pressure may lead to faster accumulation of glucose within the lens from the aqueous humor due to the increased pressure gradient, thus acceleratingcataract formation in cases of diabetes mellitus with comorbid hypertension. Regarding dyslipidemia, the initial findings of statins associated with cataractogenesis are reversed in clinical practice since a recent meta-analysis has found a clinically relevant protective effect of statins in preventing cataracts [14]. The effect is more pronounced in younger patients and with longer duration of follow-up from initial diagnosis. Large scale population studies also concur that when viewed long-term, a large population benefits from statin use with a reduced risk for cataract [15,16].

Viewed as a whole, there were significant differences between the cataract types in the total number of risk factors as well, with subcapsular and mixed type cataracts suffering from more comorbid conditions than nuclear and cortical types. The subcapsular and mixed type cataracts have the lowest incidences of cases without any comorbid conditions. The development of subcapsular cataracts, in particular, is very rare without any comorbid conditions (1.6%). These findings point to the possibility of additive interactions between the comorbid conditions that are risk factors for cataract, beyond the individual negative effects that each comorbid condition had by itself.

Among the limitations of the present study, this is a cross-sectional study of patients who already present with cataract, we cannot claim that our findings have the same validity as a longitudinal study; however, since senile cataract progresses over a very long period of time, such a study would be considerably more difficult to conduct. The relative incidence of cataract types is random within the particular population in our study, with no selection bias since they are consecutive patients and does not necessarily correspond to the general incidence of cataract types, although a lower than expected frequency of cortical and a higher than expected frequency of subcortical cataract could be attributed to causes that do not correlate to the risk factors we examined (e.g., high myopia and exposure to therapeutic doses of steroids and ionizing radiation) [17].

## 5. Conclusions

Although the present study is limited in its scope by its cross-sectional design, it presents important findings regarding the pairing of risk factors with types of cataract, and also the additive effect of diabetes mellitus, hypertension, and dyslipidemia. In the present study sample, there was a low incidence of diabetes mellitus, in contrast to hypertension, despite the large body of evidence on the correlation of diabetes mellitus with cataract.This finding raises the importance of early detection of hypertension, a cardiovascular condition that typically progresses undetected for a significant number of years. More research may be warranted in the combined effect of hypertension and diabetes mellitus in the etiopathology of cataract.

## Figures and Tables

**Table 1 medicina-55-00430-t001:** Types of cataract in the sample by gender.

	Type of Cataract				
	Subcapsular	Nuclear	Cortical	Mixed	Total
Female	74 (8.9%)	198 (23.7%)	62 (7.4%)	120 (14.4%)	454 (54.4%)
Male	54 (6.5%)	214 (25.7%)	22 (2.6%)	90 (10.8%)	380 (45.6%)
Total	128 (15.3%)	412 (49.4%)	84 (10.1%)	210 (25.2%)	834 (100%)

**Table 2 medicina-55-00430-t002:** Types of cataract in the sample by age.

	Type of Cataract				
	Subcapsular	Nuclear	Cortical	Mixed	Total
Age <65 years	18 (2.2%)	42 (5%)	8 (1%)	18 (2.2%)	86 (10.3%)
Age 65–75 years	40 (4.8%)	162 (19.4%)	36 (4.3%)	76 (9.1%)	314 (37.6%)
Age >75 years	70 (8.4%)	208 (24.9%)	40 (4.8%)	116 (13.9%)	434 (52%)

**Table 3 medicina-55-00430-t003:** Results of the ophthalmological examination by types of cataract (mean/SD).

Parameter		Type of Cataract			
		Subcapsular	Nuclear	Cortical	Mixed
BCVA (ETDRS letters)		64.1± 3.6	62.2± 2.6	63.2± 2.9	63.7± 2.7
CS (log)	with glare	0.701± 0.06	0.691± 0.02	0.251± 0.1 *	0.59± 0.04
	without glare	0.961± 0.07	0.701± 0.01	0.661± 0.08 *	0.771± 0.03
CMT (µm)		238.7 ± 12.3	245.2 ± 11.1	242.6 ± 13.1	240 ± 11.5

* statistically significant difference, SD = Standard deviation, BCVA = Mean best-corrected visual acuity, ETDRS = Early Treatment Diabetic Retinopathy Study, CS = Contrast Sensitivity, CMT = Central macular thickness.

**Table 4 medicina-55-00430-t004:** Number of individual risk factors per type of cataract.

Type of Cataract		Individual Risk Factors								Total
		AH	DM	Dysl	AH + DM	AH + Dysl	DM + Dysl	None	All	
Subcapsular	Count	56	0	0	6	34	2	2	28	128
	% within type	43.80%	0.00%	0.00%	4.70%	26.60%	1.60%	1.60%	21.90%	100.00%
Nuclear	Count	100	22	22	34	78	6	88	62	412
	% within type	24.30%	5.30%	5.30%	8.30%	18.90%	1.50%	21.40%	15.00%	100.00%
Cortical	Count	24	0	8	6	16	0	12	18	84
	% within type	28.60%	0.00%	9.50%	7.10%	19.00%	0.00%	14.30%	21.40%	100.00%
Mixed	Count	58	8	12	14	48	12	20	38	210
	% within type	27.60%	3.80%	5.70%	6.70%	22.90%	5.70%	9.50%	18.10%	100.00%
Total	Count	238	30	42	60	176	20	122	146	834
	% within type	28.50%	3.60%	5.00%	7.20%	21.10%	2.40%	14.60%	17.50%	100.00%

AH—Arterial hypertension, DM—Diabetes Mellitus, Dysl—dyslipidemia.

**Table 5 medicina-55-00430-t005:** Mean number of risk factors for each cataract type.

	*N*	Median	Range	SD
Subcapsular	128	2	3	0.813
Nuclear	412	1	3	0.982
Cortical	84	1	3	0.987
Mixed	210	2	3	0.890
Total	834	1	3	0.945

**Table 6 medicina-55-00430-t006:** Individual comparisons between cataract categories on the number of risk factors.

	Subcapsular	Nuclear	Cortical
Subcapsular	-	-	-
Nuclear	Z = 3.779, *p* < 0.001 *	-	-
Cortical	Z = 1.489, *p* = 0.136	Z = 1.396, *p* = 0.163	-
Mixed	Z = 1.097, *p* = 0.272	Z = 3.014, *p* = 0.003 *	Z = 0.645, *p* = 0.519

* statistically significant difference.

**Table 7 medicina-55-00430-t007:** The number of patients per cataract type with or without risk factors.

Type of cataract		Risk Factors		Total
		With Risk Factors	No Risk Factors	
Subcapsular	Count	126	2	128
	% within type	98.4%	1.6%	100.0%
Nuclear	Count	324	88	412
	% within type	78.6%	21.4%	100.0%
Cortical	Count	72	12	84
	% within type	85.7%	14.3%	100.0%
Mixed	Count	190	20	210
	% within type	90.5%	9.5%	100.0%
Total	Count	712	122	834
	% within type	85.4%	14.6%	100.0%

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
