# Peer review of "Hypertension is the Prominent Risk Factor in Cataract Patients"

_medicina, 2019, doi:10.3390/medicina55080430_

Round 1
Reviewer 1 Report
The authors provide an interesting article on the relationship between cataract and major cardiovascular and metabolic risk factors. Introduction provides sufficient background and includes all relevant references. The research design is appropriate and the methods are adequately described. Discussion and conclusions are adequate, as well as cited references. This manuscript is appropriate for the planned publication. Only additional editing for grammar and some language clearances, especially in abstract and results section, are required (I am not a native English speaker and I am not totally qualified for that). Also, some modifications in results section are needed. Table 1 would be much clearer if it included only the number and percentage of patients divided by gender and type of cataract; % within gender and % within type of cataract can be omitted. In Table 3 it would be better to bring up only the median and the range (minimum – maximum); mean, standard deviation and standard error values are not appropriate and can be omitted.
Author Response
Dear reviewer
Thank you for taking the time to review our manuscript. Detailed response to all points follows:
The authors provide an interesting article on the relationship between cataract and major cardiovascular and metabolic risk factors. Introduction provides sufficient background and includes all relevant references. The research design is appropriate and the methods are adequately described. Discussion and conclusions are adequate, as well as cited references. This manuscript is appropriate for the planned publication. Only additional editing for grammar and some language clearances, especially in abstract and results section, are required (I am not a native English speaker and I am not totally qualified for that).
-Thank you for your comments, the abstract and results section were rewritten in part to make the text easier to follow for the reader as it was complicated in expression and very technical in some parts.
Also, some modifications in results section are needed. Table 1 would be much clearer if it included only the number and percentage of patients divided by gender and type of cataract; % within gender and % within type of cataract can be omitted. In Table 3 it would be better to bring up only the median and the range (minimum – maximum); mean, standard deviation and standard error values are not appropriate and can be omitted.
-Thank you for your comments. Table 1 has been extensively modified to address your points as well as the requirements of Reviewer #3. An additional table (table 2) presents findings from the ophthalmological examination following the suggestion of Reviewer #2. Table 3 (now Table 4) has been modified with mean and SE of the mean removed while range has been added. We have retained standard deviation in the table since it is employed in the Kruskal-Wallis and the Mann-Whitney tests that follow that table in the manuscript.

Reviewer 2 Report
The present research presents the data showing hypertension or hypertension associated with other disease conditions such as diabetes mellitus (DM) or dyslipidemia poses a significant risk factor for developing a cataract. The present cross-sectional study has been carried out by recruiting a large group of patients with different demographic conditions.
The inclusion and exclusion criteria have been described properly. The research has been carried out following IRB protocols and appropriate signed consent has been obtained from the study participants.
The previous published paper (Yu X, Lyu D, Dong X, He J, Yao K (2014) Hypertension and Risk of Cataract: A Meta-Analysis. PLoS One 9(12): e114012. doi:10. 1371/journal.pone.0114012) on the same research area has identified arterial hypertension as one of the major risk factors for posterior subcapsular cataract (PSC) in a wide population-based Meta-analysis. Here, the present authors took an adequate approach to test the earlier research findings in a real patient group. With an extension, the present research, in fact, has carried out by considering different sub-classifications of cataract in association with either hypertension or hypertension associated with other cardiovascular disease conditions, such as diabetes mellitus (DM) or dyslipidemia.
Major comments:
1. The information provided in the experimental methods section is scanty. It has been stated that the study and classification of cataract have been done by an appropriate method (LOCS III grading system) by an independent rater. In order to explain the cataract grading system, the manuscript must provide a brief overall description of how this entire procedure has been performed.
2. Since only one independent rater was responsible for the cataract rating related data acquisition, what specific measures have been taken to address any concerns related to study bias? The lens opacity grading is subjective and there are possibilities for inclusion of potential biases.
3. The manuscript only presents data in tabular form for the cataract classifications. From a patient’s visual perspective, the authors need to show other ocular data such as the results for the visual acuity test, contrast sensitivity, slit-lamp exam and retinal exam.
4. The present research concluded that the contribution of uncomplicated diabetes mellitus (DM) is not significant in the present study population and the statement (Page 2, lines 155-156) ….”there is a possibility that hypertension may exacerbate the negative impact of diabetes mellitus in the progression of diabetic cataract.” has not explained well. How does hypertension exacerbate the negative impacts of DM?
Minor comments:
Explain the full form of LOCS III classification: Lens Opacities Classification System III
Author Response
Dear Reviewer
Thank you for taking the time to review our manuscript. A detailed response to the points you raised follows:
1. The information provided in the experimental methods section is scanty. It has been stated that the study and classification of cataract have been done by an appropriate method (LOCS III grading system) by an independent rater. In order to explain the cataract grading system, the manuscript must provide a brief overall description of how this entire procedure has been performed.
2. Since only one independent rater was responsible for the cataract rating related data acquisition, what specific measures have been taken to address any concerns related to study bias? The lens opacity grading is subjective and there are possibilities for inclusion of potential biases.
-(Answer to both points) Thank you for pointing out the misrepresentation of the procedure; all cataract cases had been previously rated with the LOCS III system by an ophthalmology trainee and an ophthalmology consultant during the scheduled outpatient appointment as is standard practice in the clinic - an independent expert in cataract (last author) confirmed that rating for the study; the latter expert's opinion prevailed in case of inter-rater disagreement. This point has been elucidated in the manuscript and a more detailed description of the LOCS III system has been included as well.
3. The manuscript only presents data in tabular form for the cataract classifications. From a patient’s visual perspective, the authors need to show other ocular data such as the results for the visual acuity test, contrast sensitivity, slit-lamp exam and retinal exam.
-Thank you for your comment, we have added a separate table (table 2) with results from best-corrected visual acuity, contrast sensitivity and cortical mean thickness. Table 1 was also remodeled following input from reviewers 1 & 3 to include age.
4. The present research concluded that the contribution of uncomplicated diabetes mellitus (DM) is not significant in the present study population and the statement (Page 2, lines 155-156) ….”there is a possibility that hypertension may exacerbate the negative impact of diabetes mellitus in the progression of diabetic cataract.” has not explained well. How does hypertension exacerbate the negative impacts of DM?
-Thank you for your point. We have added a hypothesis regarding an additive effect of hypertension on the pathophysiological process of cataract in patients with DM.
Minor comments:
Explain the full form of LOCS III classification: Lens Opacities Classification System III
-Thank you for your point, this issue has been rectified in the text

Reviewer 3 Report
The study needs to be completed by a control group. All comparaisons may result dued to age (diabetes and hypertension are Common in elderly people
Number of subjects seems to be low
Author Response
Dear Reviewer
Thank you for taking the time to review our manuscript. A detailed response to the points you raised follows:
The study needs to be completed by a control group. All comparaisons may result dued to age (diabetes and hypertension are Common in elderly people)
-Thank you for your observation; the text did not explicitly mention that we are dealing exclusively with senile cataract. This fact has been made clear in the revised manuscript. Also, Table 1 now has age frequencies for each group and the text clearly mentions that there were no statistical differences between the groups with regards to age. This is also now pointed out clearly.
Number of subjects seems to be low
-Thank you for your comment; we have added results from a power analysis on the effect size of the differences between the cataract groups on the total number of risk factors so as to better represent the clinical significance of those findings (page 5, lines 137-139). The effect sizes show that increasing the sample size would not yield any significant improvements in the results.

Round 2
Reviewer 2 Report
The authors have provided necessary answers to the comments of individual reviewers. In this regard, the inclusion of a new table (Table 2) is very important. The authors have taken adequate care in order to improve the quality of the manuscript.
I have a few minor comments:
1. Replace reference [12] by an appropriate original research article. Citation from a book is not appropriate in a research manuscript.
2. Discussion section, lines 185-186: ‘Intraocular pressure and systemic blood pressure have been found to correlate [13] Increased intraocular pressure…..’ This statement needs necessary punctuation in order to transmit the message clearly.
3. The page numbers in the present manuscript are very erratic and need to be fixed.
Author Response
Reviewer #2
The authors have provided necessary answers to the comments of individual reviewers. In this regard, the inclusion of a new table (Table 2) is very important. The authors have taken adequate care in order to improve the quality of the manuscript.
I have a few minor comments:
1. Replace reference [12] by an appropriate original research article. Citation from a book is not appropriate in a research manuscript.
- Thank you for your comment, an appropriate original article has been cited.
2. Discussion section, lines 185-186: ‘Intraocular pressure and systemic blood pressure have been found to correlate [13] Increased intraocular pressure…..’ This statement needs necessary punctuation in order to transmit the message clearly.
-Thank you for your comment, punctuation marks were missing and have been added.
3. The page numbers in the present manuscript are very erratic and need to be fixed.
-Thank you for your comment, page numbers have been fixed.
Reviewer 3 Report
in my opinion in the paper there are some Biases,
1. patient that undergoes to a cataract surgery are different from patient who has cataract. in this study we can loss a lot of informations.
2. there are data in literature on the association between Diabetes and cataract formation, in this sample there are few diabetic patients than data reported in literature may be a selection bias)
3. dyslipidemia is usually treated with statins that may cause cataract and this may cause a selection bias because cataract formation occurs earlier than patients without statins treatment. also this must be discussed
4. the number of cases is too low to allow a good stratification of data.
5. in the stratification there are too much subcapsular cataract (according literature data) the authors must discuss about it (refractive error, drug taking, traumatic events).
6. cortical number is too low, the authors must discuss about it.
7. there is a lack in control group (patients with same age and sex that do not undergo to cataract surgery)
Author Response
Reviewer #3
in my opinion in the paper there are some Biases,
1. patient that undergoes to a cataract surgery are different from patient who has cataract. in this study we can loss a lot of informations.
-thank you for your comment, as you point out this is a cross-sectional study of patients with cataract and not a longitudinal one. We cannot claim that our findings have the same validity of a longitudinal study; however, since senile cataract progresses over a very long period of time, such a study would be considerably harder to organize and fund. This sentence has been added to the manuscript in verbatim.
2. there are data in literature on the association between Diabetes and cataract formation, in this sample there are few diabetic patients than data reported in literature may be a selection bias
-thank you for your comment, this is a study of consecutive patients that met the inclusion criteria, so any variations from the reported percentages are completely random. This sentence has been added to the results
3. dyslipidemia is usually treated with statins that may cause cataract and this may cause a selection bias because cataract formation occurs earlier than patients without statins treatment. also this must be discussed
Thank you for your constructive comment. Actually results from recent meta-analyses and large-scale population surveys indicate a clinically relevant protective effect of statins in preventing cataracts, mostly in younger patients. We have added several citations to this effect and expanded on the discussion section.
4. the number of cases is too low to allow a good stratification of data.
Thank you for your comment. There are several combinations of risk factors with relatively low numbers but this is a consecutive patient population from a large tertiary clinical center serving a metropolitan area, hence there is no reason to assume large fluctuations from the reported percentages.
5. in the stratification there are too much subcapsular cataract (according literature data) the authors must discuss about it (refractive error, drug taking, traumatic events). and 6. cortical number is too low, the authors must discuss about it.
(answering both comments 5 and 6) thank you for your constructive comments, the relative sizes of the groups are random within the particular population with no selection bias since they are consecutive patients and do not reflect general incidence of cataract types; this has been clearly added in the manuscript with some speculation as to the cause of relative frequencies, as suggested by the author, with appropriate reference.
7. there is a lack in control group (patients with same age and sex that do not undergo to cataract surgery)
-Thank you for your comment, this is a within-group comparative study regarding comparison of the risk factors within the cataract groups and not a between-group cataract vs non-cataract comparison.